# Contribution of Pumping Action of Wave-Induced Pore-Pressure Response to Development of Fluid Mud Layer

**Zhongnian Yang [1], Yongmao Zhu [2], Tao Liu [2,3,\*], Zhongqiang Sun [2], Xianzhang Ling [4] and Yuqian Zheng [2]**

[1]   School of Civil Engineering, Qingdao University of Technology, Qingdao 266033, China;
     zhnyang110@163.com
[2]   Key Laboratory of Shandong Province for Marine Environment and Geological Engineering,
     Ocean University of China, Qingdao 266100, China; zhuyongmao@stu.ouc.edu.cn (Y.Z.);
     sun_zhongqiang@163.com (Z.S.); zhengyuqian@stu.ouc.edu.cn (Y.Z.)
[3]   Laboratory for Marine Geology, Qingdao National Laboratory for Maine Science and Technology,
     Qingdao 266061, China
[4]   Harbin Institute of Technology, Harbin 150001, China; ymzhuouc@163.com
\*   Correspondence: ltmilan@ouc.edu.cn; Tel.: +86-0532-6678-1020

**Abstract:** To investigate the vertical migration response of fine sediments, the pore pressure response of the silty seabed under the action of waves was tested. Under the action of waves, there is an obvious pumping phenomenon in the sludge accumulated by pore pressure. The excess pore water pressure caused by the waves in the seabed is unevenly distributed with respect to depth and there is an extreme value of up to 1.19 kPa. The pressure affects the liquefaction properties of the sludge. According to instantaneous-liquefaction judgment, the liquefaction of surface soil occurs, but the soil is not completely liquefied. Using theoretical calculations, the vertical source supply of floating mud development was analyzed. The pumping effect of the wave-induced excess pore pressure manifests in two aspects, as follows: (1) The centralized migration of splitting channels, which is visible to the naked eye, and (2) the general migration of fine particles between particle gaps at the mesoscopic level, which accounts for up to 22.2% of the migration of fine particles.

**Keywords:** pumping effect; wave-induced; silty seabed; fluid mud layer

---

## 1. Introduction

Fluid mud refers to a layer of high-concentration sediment-laden water near the seabed. It is mainly composed of clay and silt with high viscosity and the predominant particle size is generally ≤63 μm [1]. Fluid mud has a clear interface with the upper water body and has high mobility. It has a unique sediment movement pattern in muddy coastal estuaries [2,3]. This has been observed in the Thames estuary in England, the Gironde estuary in France [4], the Amazon estuary in Brazil [5,6], and the Yellow River and Yangtze River estuaries in China [7].

The liquefaction of sand and the resuspension of sediment under the action of waves have long been the focus of coastal research. Most scholars focus on liquefaction mechanisms and resuspension fluxes [8–11]. However, few researchers pay attention to the vertical pumping effect of wave-induced pore pressure from a microscopic viewpoint, particularly the development of floating mud in silty soil [12]. A large amount of fluid mud can cause ecological, environmental, and navigation safety problems, such as burying benthic organisms [13], sediment eutrophication [14], and sudden channel siltation, disturbing sounding results and affecting the judgment of the navigable water depth [15].

Additionally, its interaction with the suspended load and bedload has an important impact on the topographic and geomorphological evolution of estuary areas [16,17], such as channel siltation and even navigation safety. It can block the channel and cause ships to run aground. In July 2015, the Yangtze River Estuary waterway was blocked by typhoon Canhong, which prevented the entrance of 15 huge ships and the port production was seriously affected.

The main research methods of fluid mud are field observation, laboratory test, theoretical analysis, and numerical simulation [18]. The problem of floating mud was discovered via field observations, which employ frozen sampling, bathymetry, the gamma-ray method, the ultrasonic method, the tuning fork density method, and the coupling method [19]. The measurement principles can be categorized as direct measurement, isotope measurement, and acoustic measurement. Laboratory tests are mainly divided into two types, as follows: The experimental study of the rheological properties of the sludge [20,21] and the experimental study of the movement characteristics of the sludge under dynamic action, including wave action [22–24] and current action [25–28]. On the basis of field observations and laboratory tests, the formation conditions [29] of sludge have been investigated and the numerical simulation [30] of the sludge response under dynamic action has been performed via theoretical analysis.

In view of the development mechanism of fluid mud, the prevailing view is that the conditions for the formation of fluid mud include an abundant fine sediment supply, relatively weak hydrodynamic conditions, and appropriate salinity. Salinity and a weak dynamic force are the external conditions for the flocculation and sedimentation of fine sediment, and abundant fine sediment provides the material source for the formation and development of fluid mud. Studies have been performed on the flocculation environment of fine sediment. Through field observations, Li found that the Yangtze River estuary has a good environment for flocculation of fine sediment [31]. Li and Zhang discovered that salinity and sediment concentration significantly contribute to the flocculation of fine sediment [32]. The periodic change of the flow velocity is the most significant factor affecting the flocculation size of fine sediment [33]. Through shallow section and dual-frequency sounding, Wen considered the stratification of the floating mud in the Yellow River estuary [34].

It is generally believed that there are two forms of material supply of floating mud. The first is direct input from rivers. Winterwerp and Li reported that the potential of density inflow is a special manifestation of the rapid deposition of sediment at the bottom, which provides a rich material source for the formation of floating mud [35,36]. Second, for sediment resuspension supply, whose dynamic mechanism is that waves, currents, storms, and other comprehensive effects on the seabed, the shear stress of the wave-current combined bed is significantly increased, leading to the resuspension of fine sediment near the coast [37,38]. Some researchers use numerical simulations to study the hydrodynamic effects that are responsible for the sediment resuspension. Ranasinghe, R simulates the coastal morphology of one-way flow by using the morphodynamic model Delft3D [39]. Gallerano proposed an integral formulation for the contravariant suspended sediment advection-diffusion equation and used it for the sea-bottom dynamic simulations [40]. Some non-hydrostatic models applied to shallow coastal waters have been proposed one after another [41–43]. Both of them are based on field observations and numerical simulations. Field observation reveals from a macro perspective. The analysis of numerical simulations is based on simple conditions, such as symmetrical bed disturbance and unidirectional flow. There is a lack of dynamic process research from a micro perspective, which is insufficient to fully explain the material supply mechanism of sludge development.

In this study, silt from the Yellow River Delta was the research object. The pore pressure accumulation process of soil under wave action was analyzed by wave flume test and the instantaneous liquefaction depth was calculated. Based on the analysis of particle size composition of soil, the vertical migration of soil particles under wave action was studied, and the contribution of fine particles to surface sludge was discussed. Through theoretical calculation of different D50 particles, the source and supply mechanism of sludge development are revealed from a mesoscopic point of view, in order to supplement and enrich the development mechanism and transport law of fluid mud.

## 2. Materials and Methods

### 2.1. Experiment Introduction

A wave-tank experiment was performed at the Geotechnical Laboratory of the Ocean University of China and the experimental setup is shown in Figure 1. The size of the sink was 3 m × 0.6 m × 1.2 m and the size of the lower tank was 1 m × 0.6 m × 0.5 m. An air compressor was used to power the unit. A double-acting cylinder was installed at one end of the water tank. The cylinder can provide pressure of 0.05–1.0 Mpa and the wave-making plate reciprocated to push the water body to generate waves. The waveform output was changed by adjusting the cylinder stroke and the cylinder damping while the other end was equipped with a wedge-shaped sponge block for eliminating the influence of wave reflection and ensuring the stable output of the waveform. As shown in the figure, the soil tank was filled with saturated silt with thickness of approximately 40 cm. A kaolin layer was set at a distance of 5 cm from the surface of the soil and its thickness was <0.5 cm. Before and after the wave action, the sample was allowed to stand for more than 4 h and the surface layer was extracted. The particle-size composition was determined using a laser particle-size analyzer. In this experiment, six YY-2B pore water pressure sensors were employed, which were located 5, 10, 15, 20, 25, and 30 cm from the surface of the soil. Before being buried, the pore-pressure sensors were soaked in clean water for 24 h and continuously vibrated to ensure that the internal air was completely discharged. The YY-2B pore water pressure sensor uses the imported piezoresistive pressure sensor as the sensitive device. The water pressure acts on the pressure film of the sensor through the permeable stone, which transforms the water pressure into the output of voltage signal and connects with the secondary instrument to measure the pore water pressure.

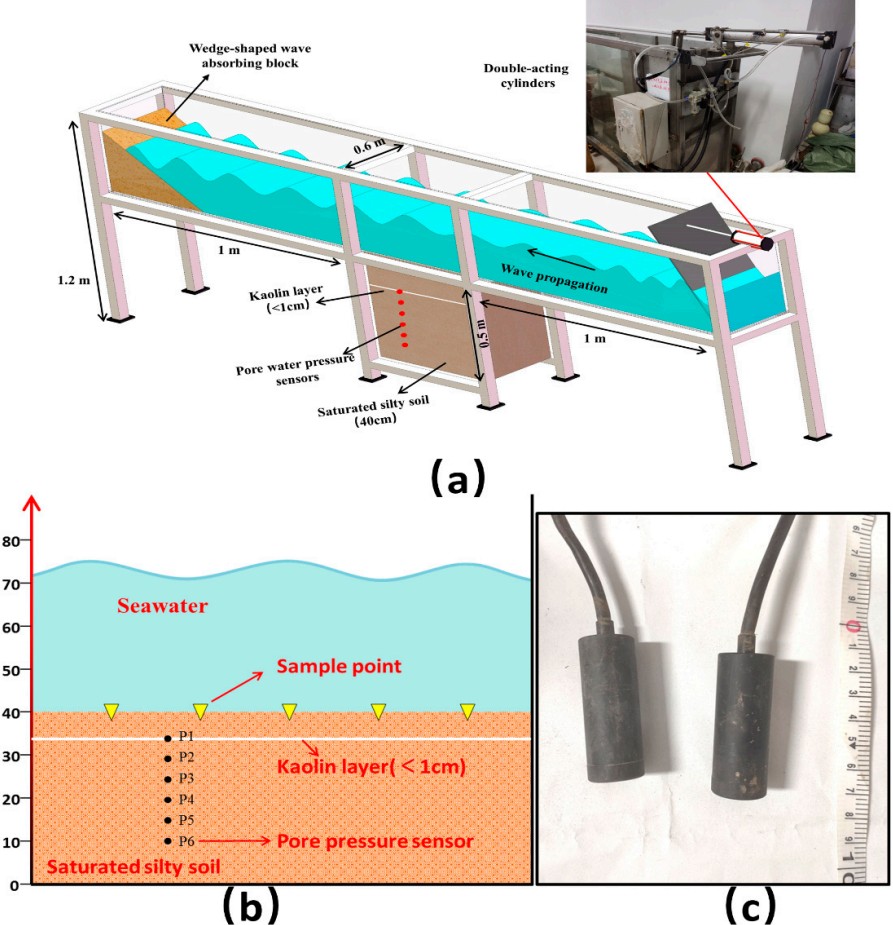

**Figure 1.** (**a**) Experimental device; (**b**) pore-pressure sensor placement; (**c**) pore-pressure sensor.

The water was artificial seawater with 35‰ salinity. The silt was taken from the tidal flat of the Yellow River estuary and had a median particle size (D50) ranging from 32 to 41 μm. The properties of the test sludge are shown in Table 1. Kaolin with 4000 meshes and D50 ranging from 2.1 to 2.8 μm was purchased from industrial factories.

**Table 1.** Properties of the test silt.

| Property | | Value |
|---|---|---|
| Moisture Content ($\omega$) | | 28.1% |
| Density ($\rho$) | | 2.0 g/cm$^3$ |
| Dry density ($\rho_d$) | | 1.55 g/cm$^3$ |
| Specific gravity ($G_s$) | | 2.7 |
| Void ratio ($e$) | | 0.74 |
| Porosity ($n$) | | 44% |
| Plasticity index ($I_p$) | | 6.9 |
| Saturated unit weight ($\gamma_s$) | | 16.0 kN/m$^3$ |
| Shear strength of uu test | Cohesion ($c$) | 7 kPa |
| | Internal friction angle ($\varphi$) | 20° |

Before the sensors were used to collect data, their sensitivity and rate were checked. The voltage values of the six pore-pressure sensors stabilized at different depths were recorded separately. According to the linear-regression method, the regression curves of the voltage values measured by the pore-pressure sensors and the true pore pressures were obtained, and the sensor calibration was completed. The results are presented in Figure 2.

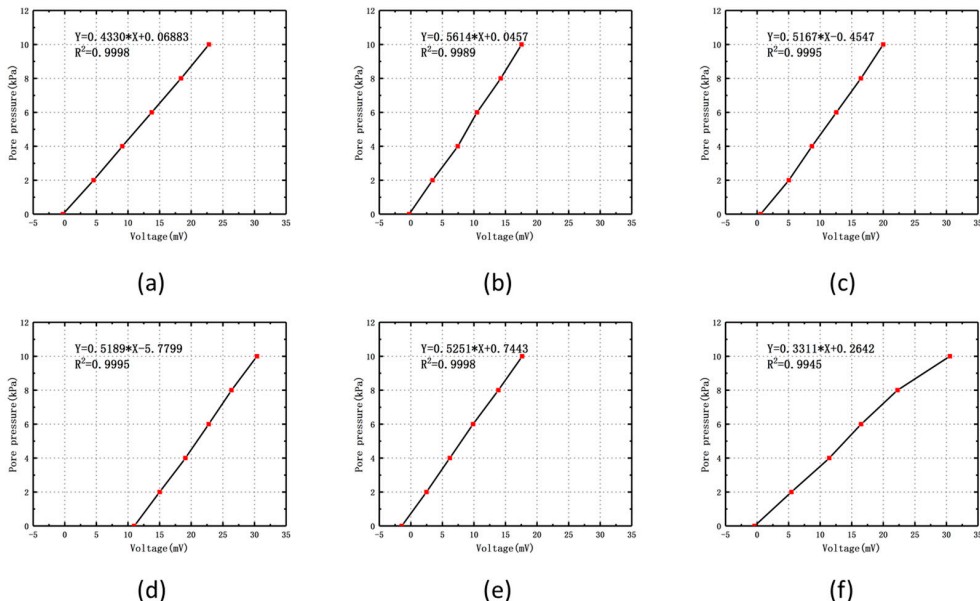

**Figure 2.** Calibration results for the pore-pressure sensors: (**a**) 5 cm from the soil surface; (**b**) 10 cm from the soil surface; (**c**) 15 cm from the soil surface; (**d**) 20 cm from the soil surface; (**e**) 25 cm from the soil surface; and (**f**) 30 cm from the soil surface.

The calibration results indicate that the pore-pressure sensor voltage display had a good linear relationship with the actual pressure value and could be used for experiments.

### 2.2. Quantification of Kaolin

The results of particle-size measurement can not only indicate parameters such as D50 and the average particle size ($D_{av}$), but also the distribution of the particle size and its cumulative distribution. By studying the grain-size characteristics of sediments, the transport mode of the sediments can be

determined and the environmental factors affecting the grain-size change of the sediments can be identified, particularly the material sources and hydrodynamic environment [44,45].

The experimental silt and kaolin were obviously different with regard to their origins and particle-size compositions. In this experiment, the silt had a larger particle size (Figure 3; D50 of approximately 32–49 μm) and was taken from the tidal flat of the Yellow River estuary. It was formed by natural accumulation after long-distance transportation and long-term screening of the water flow. It had good sorting and grinding roundness and its material was relatively consistent, approximating the normal curve. The experimental kaolin had a finer particle size (Figure 3; D50 of approximately 2.1–2.8 μm). It was manufactured in industrial batches and screened artificially, exhibited poor sorting and grinding roundness, and its particles were sharper and finer than those of the silt.

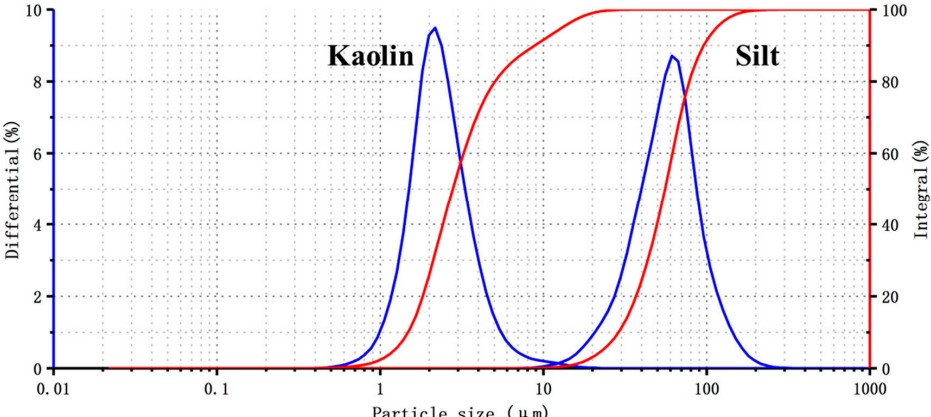

**Figure 3.** Particle-size distribution curves of the silt and kaolin used in the experiment.

Figure 4 shows the grain-size distribution curves of mixed kaolin and silt. The main component was 80% kaolin mixed with 20% silt. There are two independent peaks in the grain-size distribution curve, and the maximum values of the two peaks correspond to the grain-size ranges of kaolin (D50 = 2.1 μm) and silt (D50 = 34 μm). Therefore, the number of peaks in the particle-size distribution curve was used to qualitatively determine whether there were many components in the sample and whether the corresponding components could be qualitatively determined by the size range corresponding to the maximum value. The sample size distribution curve exhibited multiple peaks, indicating the superposition of single peaks with different particle sizes. The particle-size distribution of a component can be obtained by decomposing the particle-size distribution curve. The component proportions of different particle sizes (corresponding to different material sources) can be quantitatively assessed according to the cumulative distribution.

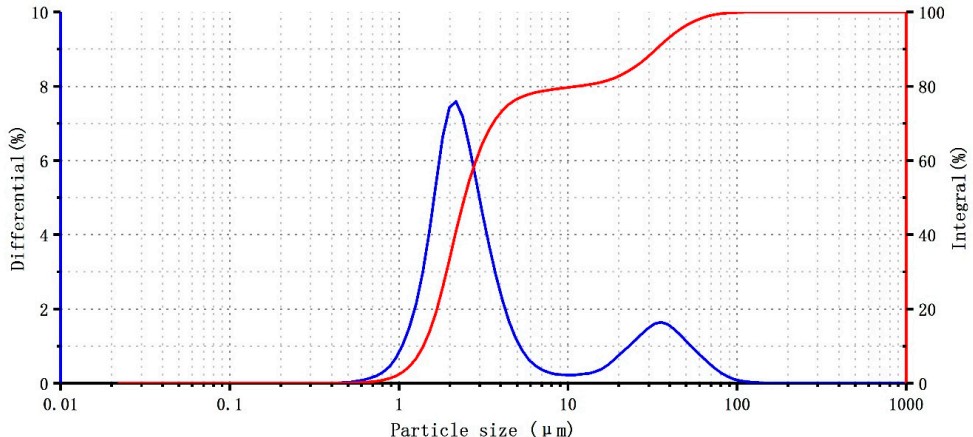

**Figure 4.** Particle-size distribution curves of mixed kaolin and silt.

## 2.3. Experimental Procedure

To simulate the physical properties of the silt under natural consolidation conditions, the soil sample was saturated and an air-dried powder sample was selected for crushing to remove impurities such as grass and branches. The sieved soil sample was placed in a blender and standard seawater was added for agitation to ensure uniform mixing of the sample. The pore water pressure sensor was pre-fixed in the middle of the soil tank at a predetermined depth to prevent disturbance to the soil after embedding. The mixture was slowly injected along the inner wall into the soil tank until the thickness was approximately 35 cm. After the soil was stable, a layer of kaolin with a thickness of <0.5 cm was evenly deposited on the surface, and then soil was added until the thickness reached 40 cm. The experimental group without the kaolin layer directly injected the mixture into the 40 cm. After the soil was stable, standard seawater was slowly injected into the pool along the inner wall until it was approximately 35 cm from the soil surface. After 12 h, the soil naturally consolidated under the hydrostatic pressure and the wave loading test was started. Before and after the wave action, four surface samples were uniformly sampled in the soil tank using a sampling tube to measure the particle size. To prevent component interference between the different experimental groups, the samples in the artificial seawater and soil tank were replaced after the end of each test.

The wave action was applied by a wave generator installed at one end of the tank to experimentally examine different wave effects (wave height and duration). The experimental group settings are presented in Table 2. Experimental groups A and C were the basic control groups, focusing on the experimental pore-pressure response and surface particle-size changes caused by the wave action. The pumping effect of the wave excess pore water pressure on the fine sediment was investigated. The pore water pressure changes for soils with different depths before and after the wave action were recorded using a YY-2B pore water pressure sensor. On this basis, the calculation of the pore-pressure accumulation caused by waves and the migration of fine particles under pumping were calculated and analyzed.

**Table 2.** Experiment setup.

| Group | Kaolin Layer | Sample Number | Wave Action | | | | Sample Number |
|---|---|---|---|---|---|---|---|
| | | | Time (min) | Wave Height (cm) | Wave Length (cm) | Wave Period (s) | |
| A | × | 1–4 | 20 | 7 | 100 | 1 | 5–8 |
| B | √ | 9–12 | 10 | 7 | 100 | 1 | 13–16 |
| C | √ | 17–20 | 20 | 7 | 100 | 1 | 21–24 |
| D | √ | 25–28 | 30 | 7 | 100 | 1 | 29–32 |
| E | √ | 33–36 | 20 | 5 | 90 | 0.9 | 37–40 |
| F | √ | 41–44 | 20 | 3 | 80 | 0.9 | 45–48 |

## 3. Results

### 3.1. Experimental Phenomena

As the wave action began, the originally consolidated soil in the trough gradually appeared to fluctuate slightly. The fluctuation period of the soil was consistent with the wave-action period and the surface layer exhibited the largest fluctuation, with an amplitude of <2 cm. The start time of the soil fluctuation was delayed with the increasing depth. The wave eventually affected the depth of the soil and increased with an increase in the wave height. The maximum depth of influence visible to the naked eye reached 15 cm below the surface layer of the soil. The overlying water body on the soil gradually became mixed with clearness. The sediment on the surface of the soil was lifted and bubbles occasionally escaped from the soil. After the wave action stopped, the fluctuation of the upper soil quickly disappeared and the soil began to be consolidated. The particles suspended in the water slowly settled and the water body recovered after standing for a long time. At the boundary of the sink, obvious small channels were observed (Figure 5a), along with splitting channels 3–17 cm in length.

Sediment deposits in the lower part of the soil followed these small channels, moving up quickly, and formed many tiny mounds on the surface of the soil. After standing for a period of time, there were numerous hilly deposits on the surface of the soil (Figure 5b). These deposits were generally <3 cm in diameter and ≤1 cm in height. The kaolin 5 cm below the surface of the soil quickly migrated along the channel to the surface layer and formed a white deposit. Groups C and D were the most obvious. Group D began with a large amount of white kaolin and then the color gradually became deeper. The soil rushed and gradually covered the white kaolin. After a long period of standing, the center of the mound-like accumulation gradually appeared to be concave. The whole process lasted approximately 1 h.

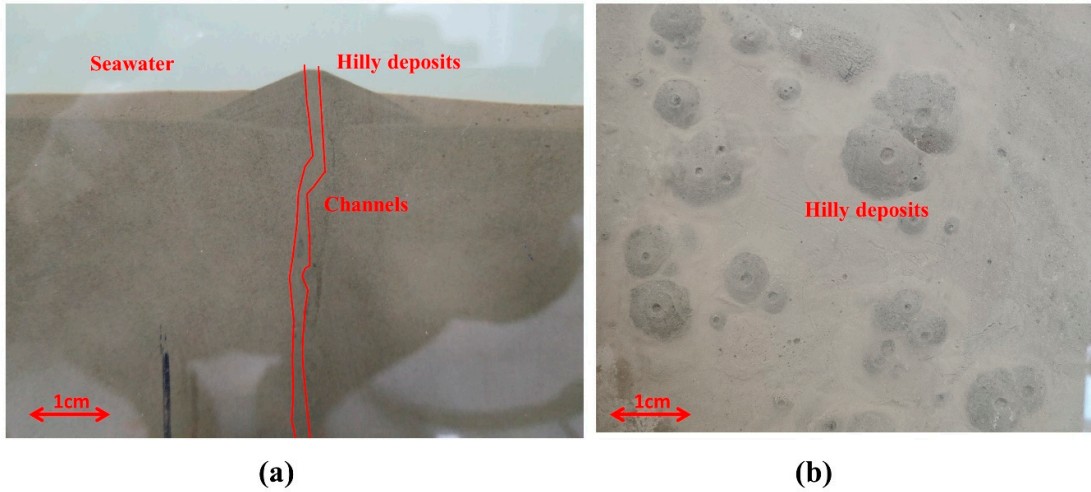

(a)                                         (b)

**Figure 5.** Upward migration and accumulation of fine particles: (**a**) Channels observed at the boundary of the flume and numerous hilly deposits; (**b**) numerous hilly deposits formed on the surface of the soil.

### 3.2. Pore Pressure Response

Figure 6 shows the pore-pressure response of the wave action (H = 7 cm, t = 30 min) 5–30 cm below the surface of the soil. The pore-pressure fluctuations of the sensors were essentially identical. Before the wave action, the pore water pressure of each layer was stable after a long period of static consolidation and the pore water pressure was equal to the overburden hydrostatic pressure. During the wave action (20–2100 s), the pore water pressure began to fluctuate with the wave load, and the pore water pressure in each layer began to accumulate (see Table 3; the maximum increment was 0.82–1.19 kPa, and the average increment was 0.76–0.94 kPa). The accumulative value and velocity of the pore water pressure in the upper layer were greater than those in the lower layer. The pore water pressure at 5 and 10 cm reached the maximum value in 150 s and then decreased slightly. The pore water pressure in each layer tended to be stable over time (they decreased by approximately 0.25–0.29 kPa). After the wave action stopped, the pore water pressure fluctuation disappeared rapidly and the pore water pressure in each layer dissipated continuously. After the wave action, the pore water pressure returned to the level observed before the wave action and remained stable for approximately 100 min.

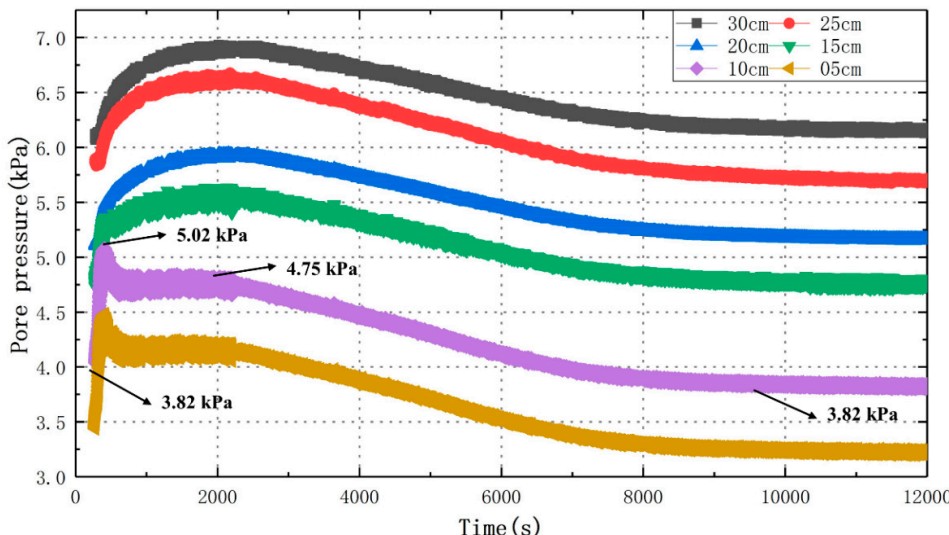

**Figure 6.** Pore-pressure responses at different depths before, during, and after wave action (H = 7 cm, t = 30 min).

**Table 3.** Analysis of the pore-pressure variation during wave action.

| Position | Before | Wave Action | | | | After |
|---|---|---|---|---|---|---|
| | Initial Pore Water Pressure $\overline{u_0}$ (kPa) | Maximum Pore Water Pressure $u_{max}$ (kPa) | Maximum Cumulative Pore Pressure $\Delta u_{max}$ (kPa) | Stable Pore Pressure $\bar{u}$ (kPa) | Stable Cumulative Pore Pressure $\overline{\Delta u}$ (kPa) | Ultimate Pore Pressure $\overline{u_1}$ (kPa) |
| 5 cm | 3.35 | 4.40 | 1.05 | 4.11 | 0.76 | 3.24 |
| 10 cm | 3.82 | 5.02 | 1.19 | 4.75 | 0.94 | 3.82 |
| 15 cm | 4.75 | 5.62 | 0.87 | 5.55 | 0.80 | 4.75 |
| 20 cm | 5.11 | 5.96 | 0.85 | 5.89 | 0.78 | 5.14 |
| 25 cm | 5.85 | 6.68 | 0.83 | 6.62 | 0.77 | 5.76 |
| 30 cm | 6.11 | 6.93 | 0.82 | 6.89 | 0.78 | 6.15 |

Note: $\Delta u_{max} = u_{max} - \overline{u_0}$, $\overline{\Delta u} = \bar{u} - \overline{u_0}$.

### 3.3. Particle Size and Kaolin Content

Table 4 presents the changes in the average particle size and kaolin content of the soil surface after the wave action. The surface particle size of the soil was generally smaller after the wave action and the kaolin was generally detected in the surface layer, except for groups A and F. A wave 3 cm in height did not cause the kaolin to move up the surface layer. The average particle size of the sample decreased with an increase in the wave-action intensity (groups D, E, and F). With the increasing wave-action time, the kaolin content in the surface layer first increased and then decreased (groups B, C, and D). The decrease in the wave height exhibited a decreasing trend (groups C, E, and F).

**Table 4.** Changes in the particle size and kaolin content after wave action.

| Group | Average Size of the Surface Particles before Wave Action $D_{av0}$ (μm) | Average Size of the Surface Particles after Wave Action $D_{av1}$ (μm) | Kaolin Content after Wave Action (%) |
|---|---|---|---|
| A | 44.9 | 36.5 | 0.0 |
| B | 33.6 | 27.6 | 3.9 |
| C | 39.0 | 16.1 | 22.2 |
| D | 44.0 | 18.6 | 11.2 |
| E | 44.4 | 19.8 | 4.5 |
| F | 41.2 | 34.0 | 0.0 |

## 4. Discussion

### 4.1. Pore-Pressure Response Analysis

A wave acting on a seabed causes an excess pore water pressure. The variation of the pore pressure in the soil and the amplitude are the factors that affect the liquefaction of the seabed soil. The data collected by the pore-pressure sensors in real time during the test represented the total pore water pressure, which comprised two parts, as follows: The pore water pressure and the static pore pressure, which does not work on the consolidation deformation and instability of the seabed soil. It is the excess pore water pressure, which is produced by the periodic loading of waves and can accumulate and dissipate in the soil.

The presence of excess pore pressure can reduce the effective stress of the seabed soil and affect the stability of the seabed. The collected data were processed, the hydrostatic pressure was subtracted from the total pore water pressure, and the excess pore pressure was analyzed.

Figure 7 presents the excess pore pressure data. Figure 7a indicates the variation of excess pore pressure with time at different depths. Nina Stark conducted a pore pressure observation at 5 cm and 20 cm depth in the intertidal zone of Cannon Beach in Yakutat, Alaska [46]. At a sediment depth of 20 cm, the period averaged pore pressure occasionally exceeded the initial mean effective stress and, at a sediment depth of 5 cm, the period averaged pore pressure exceeded the initial mean effective stress frequently. In our laboratory tests, at 5 cm and 20 cm, the periodic average pore pressure often exceeded the initial average effective stress. This may be related to the setting of the indoor test parameters, or it may be related to the characteristics of the liquefaction of the Yellow River estuary. The measurements at 20 cm sediment depth indicated that pore pressure build-up was particularly susceptible to maximum wave height. We also got similar results. This highlights the important role of wave height in pore pressure accumulation. As shown in Figure 7b, the pore water pressure was unevenly distributed with respect to the depth and there was an extreme value. The maximum value occurred 10 cm below the seabed; the pressure gradually decreased with the increasing depth. The reason for this phenomenon is that the super-porosity generated by the wave cyclic load in the seabed soil was a dynamic pore pressure, which accumulated and dissipated over time in the seabed soil. Although the wave acted on the surface of the seabed to produce an instantaneous maximum excessive pore pressure, the pore pressure in the soil near the surface of the seabed dissipated quickly, often without having to accumulate [47–50]. Therefore, the super-pore pressure was not maximized at the surface of the seabed; rather, it gradually accumulated with the increasing depth along the surface layer, reached a maximum at a certain depth from the surface layer, and (because the wave acted on the soil) gradually decreased with the increasing depth. Thus, at a certain depth, the pore pressure was gradually attenuated.

### 4.2. Discrimination of Instantaneous Liquefaction

The criterion of instantaneous liquefaction under wave action was used to compare the excess pore water pressure between the particles and the effective stress of the floating soil. Reaching the instantaneous-liquefaction standard only represents the trend of liquefaction at a certain time and does not indicate the liquefaction of the soil.

When the maximum excess pore water pressure $U_{max}$ generated by the wave in the soil layer was equal to the overlying effective self-weight stress $\sigma\prime$ of the corresponding soil layer, the soil layer was in the limit equilibrium state. At $U_{max} \leq \sigma\prime$, the soil body was stable, and at $U_{max} > \sigma\prime$, the soil layer was liquefied.

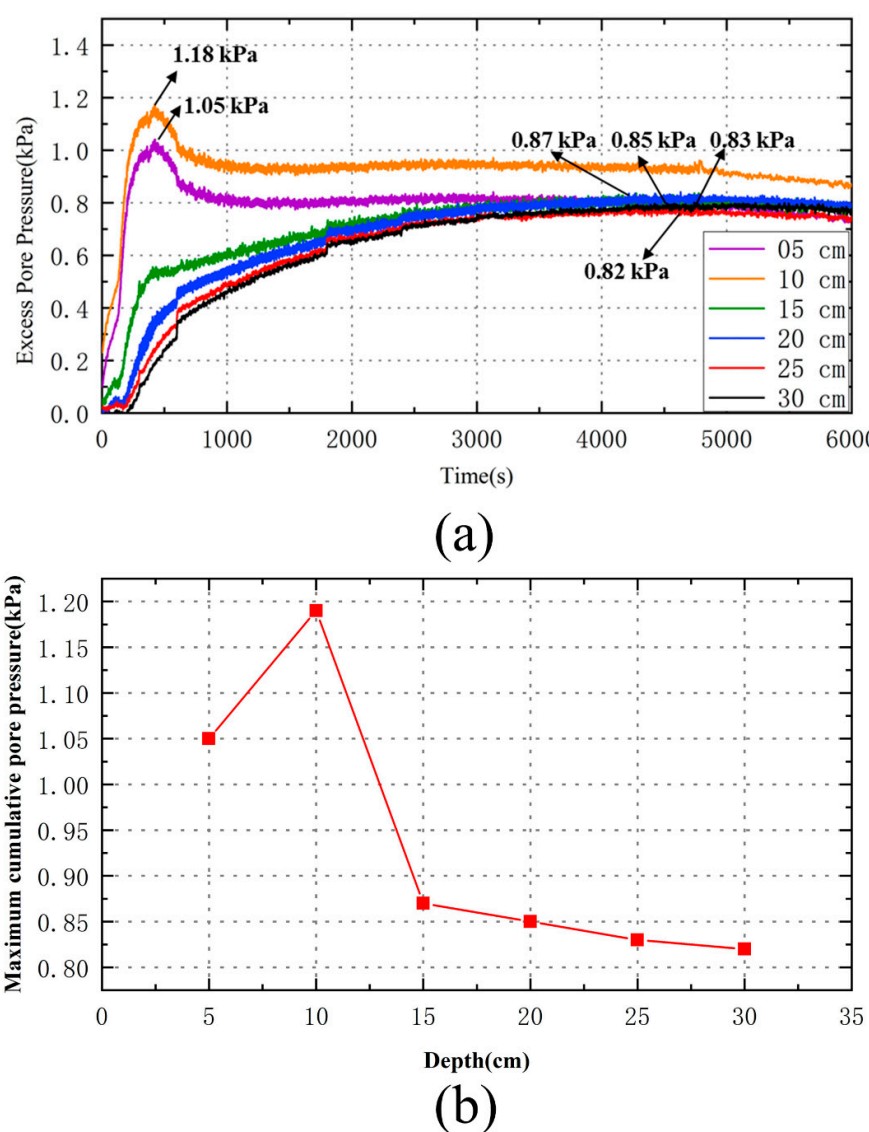

**Figure 7.** Excess pore pressure data. (**a**) Variation of excess pore pressure with time at different depths. (**b**) Schematic diagram of maximum cumulative pore pressure at different depths.

It was assumed that the soil skeleton was elastic, the pore water was incompressible, and the water was laminar in the pores. The transient response of the soil layer to the resonant wave load was analyzed. According to Yamomoto [51], a simple expression for the excess pore water pressure decay with respect to the depth was obtained, as follows:

$$U_{\max} = \frac{\gamma_w H}{2 \cosh(\lambda d)} e^{-\lambda z} \tag{1}$$

where $z$ represents the soil thickness (m), $\lambda$ represents the wavenumber (m$^{-1}$), $\lambda = \frac{2\pi}{L}$ (here, $L$ represents the wave length (m)), $\gamma_w$ represents the seawater gravity (kN/m$^3$), $H$ represents the wave height (m), and $d$ represents the water depth (m).

The effective self-weight stress of the corresponding soil layer is expressed as $\sigma\prime = \gamma\prime z$, where $\gamma\prime$ represents the submerged specific weight of the soil (N/m$^3$). By comparing the maximum excess pore water pressure, $U_{\max}$, with the overlying effective self-weight stress, $\sigma\prime$, of the corresponding soil layer, the possibility of liquefaction at depth $z$ can be judged.

By considering the effect of the lateral-pressure coefficient [52], the following expression is obtained:

$$\frac{\gamma_w H}{2\cosh(\lambda d)}e^{-\lambda z} = \frac{1+2K_0}{3}\gamma'z \tag{2}$$

Through instantaneous-liquefaction discrimination calculations, the liquefaction depth was determined for each group, as shown in Table 5. The average liquefaction location of the soil was ≤5 cm. A smaller wave height yielded a lower liquefaction level. The cumulative response curve of the pore pressure under wave action indicated that although the cumulative pore pressure at the 5 cm horizon decreased, the cumulative excess pore pressure did not decrease to 0 instantaneously, suggesting that the liquefaction of this horizon occurred but that the soul was not completely liquefied.

**Table 5.** Liquefaction depth for each group.

| Group | Depth of Water (m) | Wave Height (m) | Wave Length (m) | Liquefaction Critical Thickness (m) |
|-------|--------------------|-----------------|------------------|--------------------------------------|
| A | 0.33 | 0.07 | 1 | 0.04 |
| B | 0.33 | 0.07 | 1 | 0.04 |
| C | 0.33 | 0.07 | 1 | 0.04 |
| D | 0.33 | 0.07 | 1 | 0.04 |
| E | 0.33 | 0.05 | 0.9 | 0.03 |
| F | 0.33 | 0.03 | 0.8 | 0.02 |

### 4.3. Centralized Pumping Migration Based on Splitting Channels

During the experiment, a series of splitting channels were observed in the soil after the wave action. Driven by wave-induced excess pore pressure, particles in the lower layer moved rapidly along these channels to the surface and accumulated near the channel outlet. The upwelling upstream particles in group C were collected and analyzed. The results indicated that the upwelling particles were smaller than the other particles, having an average size of only 9.5 μm. They were mainly composed of kaolin and fine silt, and the kaolin accounted for approximately 56.4% (Figure 8).

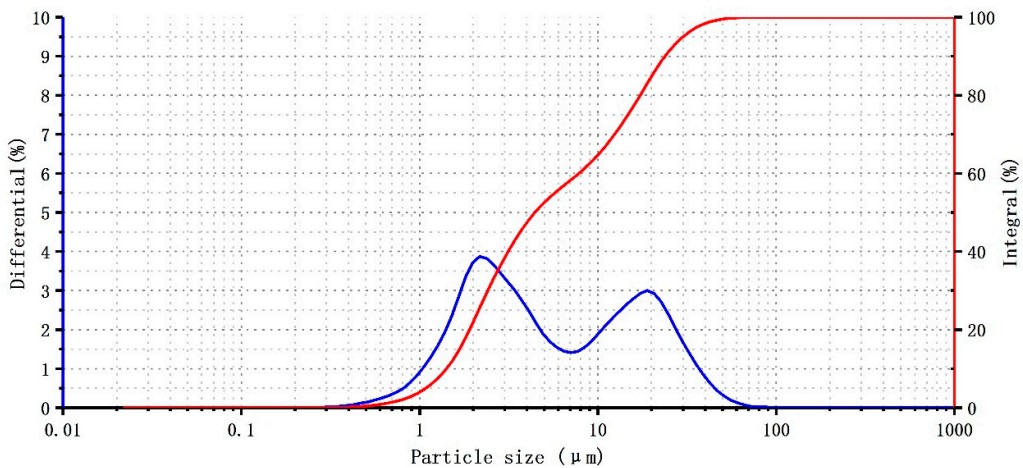

**Figure 8.** Size-distribution curve of the upwelling soil particles in channels.

Since the permeability of the silt was relatively poor (according to experimental measurements, the permeability of the sample used was approximately $1 \times 10^{-8}$ m/s), after the wave action was stopped, the accumulated excess pore pressure of the silt could not be quickly dissipated. Additionally, owing to the significant decrease in the strength of the soil, the accumulated excess pore pressure was rapidly dissipated along the weak points (or weak paths) of the soil layer through the destruction of the soil layer, forming a series of splitting channels. As the excess pore pressure dissipated, the

liquefied soil particles in the fluid state in the soil separated from the soil skeleton. Under the action of the penetrating force, the particles moved rapidly along the channels to the surface. The fine particles were more likely to move.

Generally, a longer wave-action time (groups B, C, and D) corresponded to a larger decrease in the soil strength, the formation of a larger number of upwelling channels, and a higher kaolin content detectable on the surface layer of the soil. In the experiment, the D-channel at the beginning of the group D was obviously white. Then, the color gradually deepened and transitioned to the color of the silt, and the continuous upwelling did not end. This indicates that the initial particle granulation was dominated by kaolin with a small particle size. As the upwelling continued, the amount of kaolin near the channel gradually decreased (the kaolin thickness was <0.5 cm). The upwelling granules became mainly silt, and a large amount of silt covering the kaolin made the kaolin content relatively low. Generally speaking, in a certain period of time, the smaller the wavelength of wave action, the smaller the degree of wave action on the bottom kaolin, the fewer upwelling channels formed, the smaller the amount of fine particles upwelling, and the less kaolin content can be monitored on the surface of the soil.

### 4.4. Particle-Force Analysis

For the upper soil (0–15 cm) with full wave action, without considering the interaction between particles, the vertical forces of particles under the wave action mainly included the effective gravity of the particles, the uplift force, the seepage force, the cohesion force, and additional pressure of the membrane water (Figure 9).

(1) Effective gravity $F_g$, i.e., sediment gravity subtracted by buoyancy;

$$F_g = (\rho_s - \rho_w)g\frac{\pi}{6}d_{50}^3 \tag{3}$$

(2) Uplift force $F_y$, which can be expressed as follows according to Cao [53]:

$$F_y = \left[\frac{1}{4.684(L/h)^{0.322}}\right]^2 \frac{1}{2}\rho_w\frac{\pi}{4}d_{50}^2\frac{\pi^2 H^2}{T^2\sinh^2(kh)}\cos^2(kx - \omega t) \tag{4}$$

(3) Seepage force $F_s$, which can be expressed as follows according to Mei [54] and Xia [55]:

$$F_s = \frac{n}{1-n}\frac{m}{m+1}\frac{\pi d_{50}^3}{6}\frac{\rho\omega gH}{2\cosh(kh)}\frac{1}{\delta}\sin(kx - \omega t + \frac{\pi}{4}). \tag{5}$$

(4) Bonding force and additional pressure of membrane water, which can be expressed as follows according to Mei and Cao:

$$N_1 = \varphi\frac{\pi}{2}\varepsilon d_{50}.N_2 = \varphi\rho_w h\frac{\pi}{2}d_{50}\alpha. \tag{6}$$

Here, $\rho_s$ = 2650 kg/m$^3$ represents the density of soil particles, $\rho_w$ = 1030 kg/m$^3$ represents the density of water, g = 9.8 N/kg represents the gravitational acceleration, $H$ = 0.33 m represents the water depth, $T$ = 1 s represents the wave period, $H$ = 0.07 m represents the wave height, $L$ = 1 m represents the wave length, $k$ = 1/$L$ represents the wavenumber, $\omega$ represents the angular frequency, $\delta$ represents the thickness of the seabed boundary layer, $\delta = \sqrt{\frac{k_d G}{\omega(\frac{nG}{\beta} + \frac{1-2v}{2(1-v)})}}$, where $n$ = 0.4 represents the porosity, $\beta$ = 1.0 × 10$^7$ N/m$^2$ represents the bulk elastic modulus, $G$ = 5.0 × 10$^6$ N/m$^2$ represents the shear modulus, $v$ = 0.33 represents the Poisson's ratio, and $k_d = \frac{k_s}{\rho\omega g}$ (with $k_s$ = 1.0 × 10$^{-8}$ m/s) is the permeability coefficient, $m = \frac{nG}{(1-2v)\beta}$ presents the relative compressibility of water and the soil skeleton, $\varphi = \frac{1}{16}$ is the correction coefficient, $\varepsilon = \frac{\varepsilon_k}{\rho_w}$ is the coefficient of cohesion, $\varepsilon_k$= 2.56 × 10$^{-2}$ m/s$^2$, and $\alpha$= 0.213 × 10$^{-6}$ m is the characteristic thickness related to the size of the sand gap.

Using the foregoing formulas, the forces acting on the D50 = 3 μm, 30 μm, and 300 μm particles were calculated, as shown in Figure 9. For soil particles with D50 = 3 μm, the vertical movement is mainly affected by the uplift force, cohesive force, and the additional pressure of membrane water. For soil particles with D50 = 30 μm, the vertical movement is mainly affected by the uplift force and the seepage force. For soil particles with D50 = 300 μm, the vertical movement is mainly affected by the effective gravity, uplift force, and seepage force. For particles with a relatively small size (D50 = 3 or 30 μm), the resultant force (positive orientation) is positive with time integration (impulse), indicating that the momentum of the particles increases with time. The impulse of particles with a relatively large size (D50 = 300 μm) is negative (Figure 9).

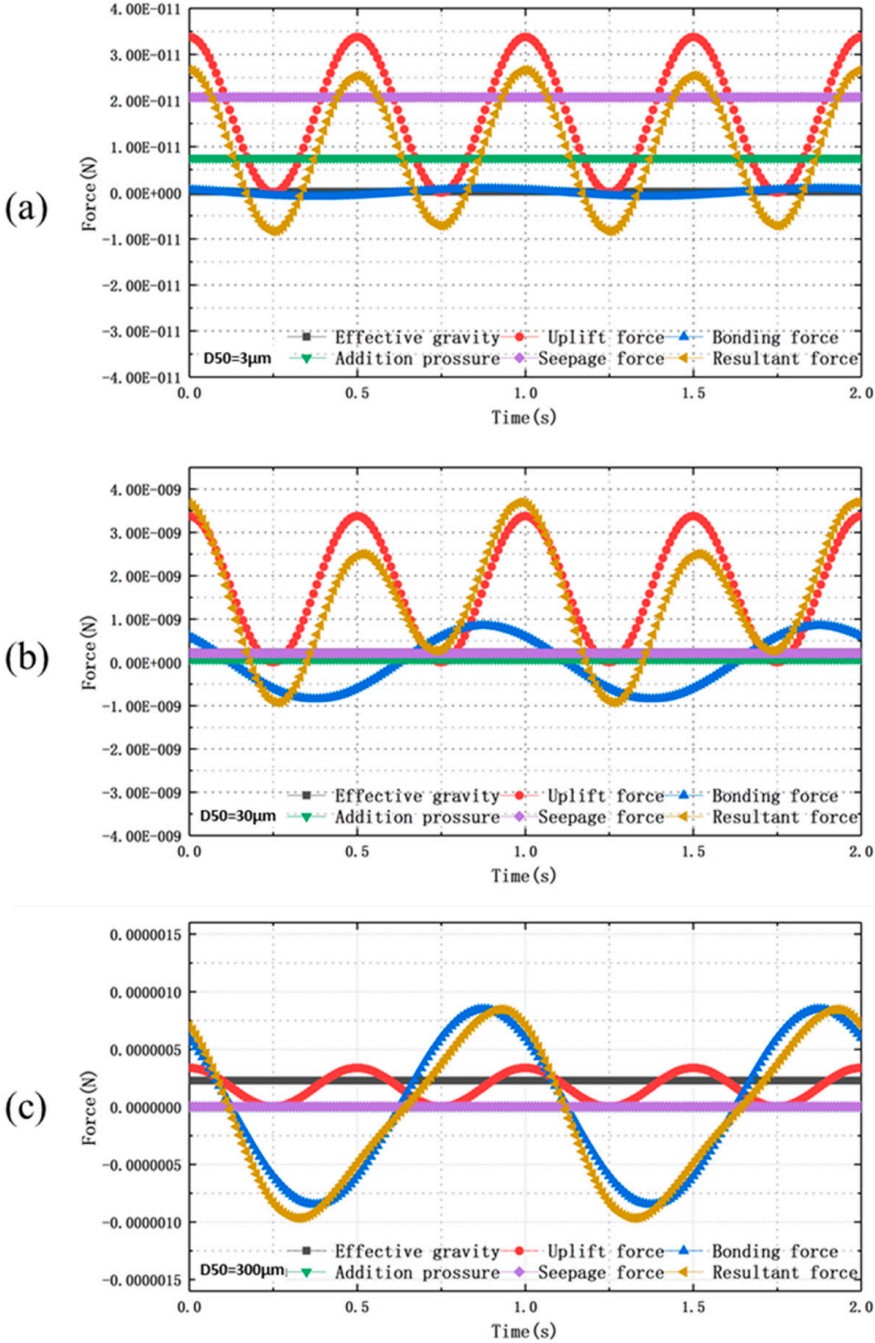

**Figure 9.** Force changes of particles under wave action in the vertical direction. (**a**) D50 = 3 μm; (**b**) D50 = 30 μm; and (**c**) D50 = 300 μm.

In summary, the pumping effect of the wave-induced excess pore pressure is mainly manifested in two aspects, as follows (Figure 10): (1) The visible centralized migration of splitting channels and (2) the general migration of fine particles between granular cracks on the mesoscale.

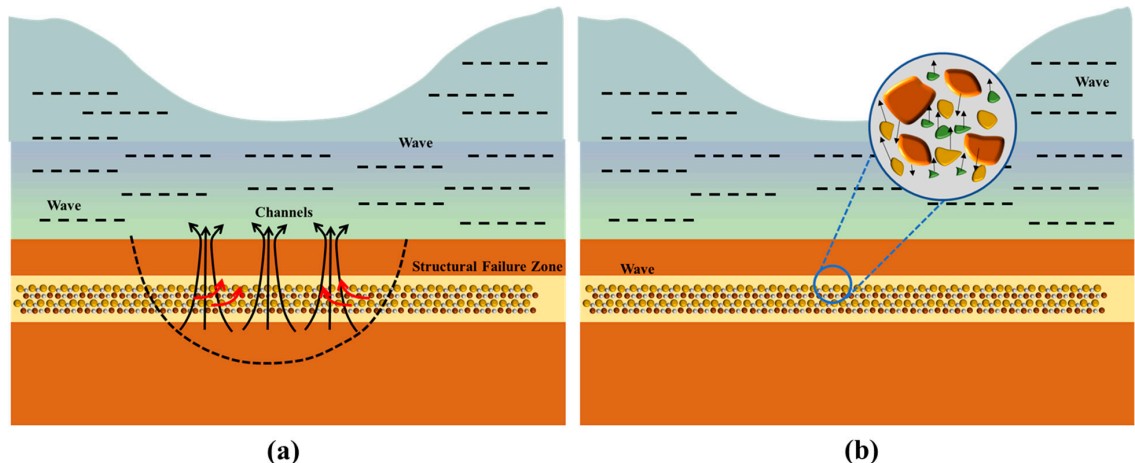

**Figure 10.** Two types of pumping effects. (**a**) Fine particles migrate in the mud diapir; (**b**) fine particles migrate between gaps.

## 5. Conclusions

The silt in the Yellow River Delta was taken as the research object and the main components of the floating mud were simulated with 4000 mesh kaolin. The responses of the pore-pressure accumulation, soil liquefaction, and particle movement under wave action were experimentally investigated. According to the experimental results and calculations, the following conclusions were drawn.

(1) Resuspension of surface soil particles occurred owing to wave action and the "pumping" effect of the wave-induced pore pressure transferred the fine kaolin from the lower layer to the surface layer, resulting in a reduced surface soil particle size after the wave action.

(2) Under the action of waves, the excess pore pressure gradually accumulated with the increasing depth along the surface layer, reaching its maximum value (1.19 kPa) 10 cm from the surface layer. The excess pore pressure gradually decreased as the depth increased further.

(3) The silt particle size was small. The cohesive force between the particles and the additional pressure of the film water had significant effects on the liquefaction characteristics of the silt. The strength of the surface soil decreased significantly. According to instantaneous-liquefaction judgment, the liquefaction of the surface soil occurred, but the surface soil was not completely liquefied.

(4) The pumping effect of the wave-induced excess pore pressure was manifested in two aspects, as follows: The centralized migration of splitting channels visible to the naked eye and the general migration of fine particles between particle gaps at the mesoscopic level, which contributed up to 22.2% of the migration of fine particles.

The accumulation of pore pressure and the upward movement of fine particles are complex. The mechanism of formation and migration of floating mud is still unclear. More studies are needed, especially with field data in order to study the detailed interaction of these processes and their correlation with beach evolution. Despite the small size of the sensors (~6 cm in length and ~2.5 cm in diameter), the sediment was disturbed by the introduction of the sensors, and certainly for the beginning of the measurements. This research for this paper was carried out under ideal regular wave conditions, which is quite different from the complex marine dynamic environment. We will further carry out in-situ testing, especially with field data to study the detailed interaction of these processes, and calibrate and verify them with laboratory tests.

**Author Contributions:** Conceptualization, Z.Y.; formal analysis, Y.Z. and Z.S.; methodology, T.L.; software, Y.Z. and Y.Z.; validation, X.L. and Z.S.; writing-original draft, Z.Y.; writing-review & editing, T.L. and Z.Y.

**Funding:** This study was supported by the National Natural Science Foundation of China (Project Nos. 41602318 and 41672272), the Laboratory for Marine Geology, Qingdao National Laboratory for Marine Science and Technology, grant No. MGQNLM-KF201710, the Fundamental Research Funds for the Central Universities, grant No. 201962011, the National Key Research and Development Program (2017YFC0307701), and the foundation for basic research of the Ocean University of China (Grant No. 201861041). We thank the Geotechnical Laboratory of the Ocean University of China for providing equipment for the experiment.

**Acknowledgments:** We appreciate the anonymous reviewers who provided comments for revising the paper.

**Conflicts of Interest:** The authors declare no conflicts of interest.

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
