# Peer review of "Contribution of Pumping Action of Wave-Induced Pore-Pressure Response to Development of Fluid Mud Layer"

_jmse, doi:10.3390/jmse7090317_

Round 1
Reviewer 1 Report
Overall, the manuscript and work are interesting, and I find it deserve publishing after addressing some comments. I added comments and questions to the pdf file which I think the authors should address before the final acceptance for the manuscript. The main points are as follow:
The paper has many assumptions that need to be supported by references or explanations. The soil tests like particle size and strength parameter tests are not discussed and only the values are shown. They need to be supported by references or tests performed. The discussion lacks supporting it with comparisons and similar studies. It should be taken to consideration that there are many publications that discuss the pore pressure response under wave action. The authors should compare their findings to such publications. There are minor editorial errors. The author needs to take a thorough look at the manuscript and fix such errors.

Reviewer 2 Report
General comments
The paper regards an experimental study of the hydrodynamic effects produced by the wave action on the pore pressure within the sediment layer of a wave tank and the vertical migration of fine sediments.
The experimental methodology carried out for this study is well described and the results are exhaustive. However, in the introduction some discussion and reference should be insert, in order to frame the physical phenomena and provide the reader with a complete overview of the alternative methodologies present in the literature.
Specific comments
In the introduction, on page 2, from line 63, there is a description of two modalities by which the sediment can be put in suspension and carried by the fluid phase. In the description of the second form of resuspension (the resuspension supply of storm events) the authors should refer to some paper where this form has been well described along with the effects produced by the combined effects of wave and currents on the nearshore sediment transport (see for example Ranasinghe et al. 2011, Gallerano et al. 2017a).
In the introduction, on page 2, from line 69, in the discussion about the hydrodynamic effects that are responsible of the sediment resuspension, some reference is missing about alternative approaches like the one based on the fully three-dimensional numerical models for the simulation of flow velocity fields induced by the wave motion (see for, example, Ma et al. 2012, Bradford 2011, Gallerano et al. 2017b).
references
Ranasinghe, R., Swinkels, C., Luijendijk, A., Roelvink, D., Bosboom, J., Stive, M., Walstra, D. (2011). Morphodynamic upscaling with the MORFAC approach: Dependencies and sensitivities. Coastal Engineering, 58(8), 806–811.
Gallerano, F., Cannata, G., Scarpone, S. (2017a). Bottom changes in coastal areas with complex shorelines. Engineering Applications of Computational Fluid Mechanics, 11(1), 396–416.
Bradford, S.F. (2011). Non-hydrostatic model for surf zone simulation. Journal of Waterway, Port, Coastal, and Ocean Engineering, 137, 163–174.
Ma, G., Shi, F., Kirby, J.T. (2012). Shock-capturing non-hydrostatic model for fully dispersive surface wave processes. Ocean Modelling, 4344, 22–35.
Gallerano, F., Cannata, G., Lasaponara, F., Petrelli, C. (2017b). A new three-dimensional finite-volume non-hydrostatic shock-capturing model for free surface flow. Journal of Hydrodynamics, 29(4):552-566.
